# Electronic ferroelectricity in monolayer graphene moiré superlattices

Le Zhang [1,2], Jing Ding [1,2], Hanxiao Xiang[1,2], Naitian Liu[1,2], Wenqiang Zhou [1,2], Linfeng Wu[1,2], Na Xin [3] ✉, Kenji Watanabe [4], Takashi Taniguchi [5] & Shuigang Xu [1,2] ✉

Extending ferroelectric materials to two-dimensional limit provides versatile applications for the development of next-generation nonvolatile devices. Conventional ferroelectricity requires materials consisting of at least two constituent elements associated with polar crystalline structures. Monolayer graphene as an elementary two-dimensional material unlikely exhibits ferroelectric order due to its highly centrosymmetric hexagonal lattices. Here, we report the observations of electronic ferroelectricity in monolayer graphene by introducing asymmetric moiré superlattice at the graphene/h-BN interface, in which the electric polarization stems from electron-hole dipoles. The polarization switching is probed through the measurements of itinerant Hall carrier density up to room temperature, manifesting as standard polarization-electric field hysteresis loops. We find ferroelectricity in graphene moiré systems exhibits generally similar characteristics in monolayer, bilayer, and trilayer graphene, which indicates layer polarization is not essential to observe the ferroelectricity. Furthermore, we demonstrate the applications of this ferroelectric moiré structures in multi-state nonvolatile data storage with high retention and the emulation of versatile synaptic behaviors. Our work not only provides insights into the fundamental understanding of ferroelectricity, but also demonstrates the potential of graphene for high-speed and multi-state nonvolatile memory applications.

Ferroelectric materials possess electrically switchable spontaneous polarizations, which offer fascinating applications for nonvolatile memories, electric sensors, radio frequency, beyond Boltzmann tyranny transistors, and synaptic devices[1–4]. In conventional ferroelectric materials, the spontaneous electric polarization is formed by the spatial separation of the cation and anion, which is switchable by an external electric field ($E$) via small lattice displacements inside a unit cell. Recently, the emergence of two-dimensional ferroelectricity not only provides the opportunity for realizing the miniaturization and

multifunction of nonvolatile devices but also opens the door to the discovery of novel ferroelectricity[5–9]. The reduced dimensionality and designable interlayer stacking in two-dimensional ferroelectric materials enable many unconventional properties different from traditional three-dimensional counterparts[10–15]. Among various two-dimensional ferroelectricity, the interfacial ferroelectricity is particularly intriguing because it arises from stacking non-polar constituents and exhibits high tunability and room-temperature functionality[16–23]. Up to now, there are mainly two kinds of interfacial ferroelectricity discovered in

[1]Key Laboratory for Quantum Materials of Zhejiang Province, Department of Physics, School of Science, Westlake University, 18 Shilongshan Road, Hangzhou 310024 Zhejiang Province, China. [2]Institute of Natural Sciences, Westlake Institute for Advanced Study, 18 Shilongshan Road, Hangzhou 310024 Zhejiang Province, China. [3]Department of Chemistry, Zhejiang University, Hangzhou 310058, China. [4]Research Center for Electronic and Optical Materials, National Institute for Materials Science, 1-1 Namiki, Tsukuba 305-0044, Japan. [5]Research Center for Materials Nanoarchitectonics, National Institute for Materials Science, 1-1 Namiki, Tsukuba 305-0044, Japan. ✉e-mail: na.xin@zju.edu.cn; xushuigang@westlake.edu.cn

two-dimensional materials: one is the sliding ferroelectricity, where the out-of-plane electric polarization can be switched by in-plane interlayer sliding benefiting from the weak interlayer van der Waals force[17–24]; the other is the unconventional ferroelectricity observed in Bernal-stacked bilayer graphene/h-BN moiré superlattice. In contrast to lattice-driven polarization, the ferroelectricity in bilayer graphene is believed to arise from spontaneous electronic polarization[4,16,25,26], which provides promising applications in ultrafast switchable memories, multiple state storage, and low-power neuromorphic devices[4,27]. Nevertheless, the two-dimensional materials exhibiting electronic ferroelectricity remain extremely rare, limited to bilayer graphene structures with specific alignment with h-BN[16,25,26,28,29].

The origin of unconventional ferroelectricity in bilayer graphene is still elusive. Previous understanding suggested it is highly related to layer-polarized flat moiré bands and tunable quadratic bands of bilayer graphene, as evidenced by the accompanying layer-specific anomalous screening effect[16,25,26,30]. The switchable electronic states are presumed to arise from layer polarization of charges and interlayer charge transfer between the top and bottom layers. This interlayer charge transfer model is based on the strong electron-electron interactions in the moiré band.

Here we report the observations of unexpected electronic ferroelectricity in monolayer graphene moiré superlattices, where the layer polarization is essentially absent and linear Dirac band weakens the electron-electron interactions, in contrast to bilayer graphene systems. However, we find that in monolayer graphene, the ferroelectricity, as well as gate-specific anomalous screening (GSAS) effect,

basically resembles that in bilayer graphene. The ferroelectricity in monolayer graphene manifests as the standard polarization-electric field ($P_{2D} - E$) hysteresis loops and atypical multiple-state switching. Our results argue that layer polarization is not an essential factor for the observation of electronic ferroelectricity in graphene/h-BN superlattices. The underlying mechanism of electronic dynamics in these systems is unveiled by performing a series of $P_{2D} - E$ loop measurements. Our observations establish graphene as the thinnest ferroelectric material known to exist, enriching the fascinating properties of this wonder material. Furthermore, our discovery will promote the applications of graphene in multi-nonvolatile switchable devices with ultrahigh mobility.

## Results

### Ferroelectric hysteresis

Our high-quality monolayer graphene devices were made from h-BN encapsulated structures with dual-gate configuration as depicted in Fig. 1a, which allows us to independently tune externally injected total carrier density $n_{total}$ and the out-of-plane displacement field $D$. To create the moiré superlattices, we intentionally aligned the straight edges of graphene with those of both top and bottom h-BN during the assembly process. Raman spectra and second harmonic generation identified the single alignment configuration, namely, graphene crystallographically aligned with the top h-BN and misaligned with the bottom h-BN by 30° (see Methods and Supplementary Fig. 1). The single alignment can be further confirmed from the electronic transport behavior as shown in Fig. 1b, which exhibits typical graphene

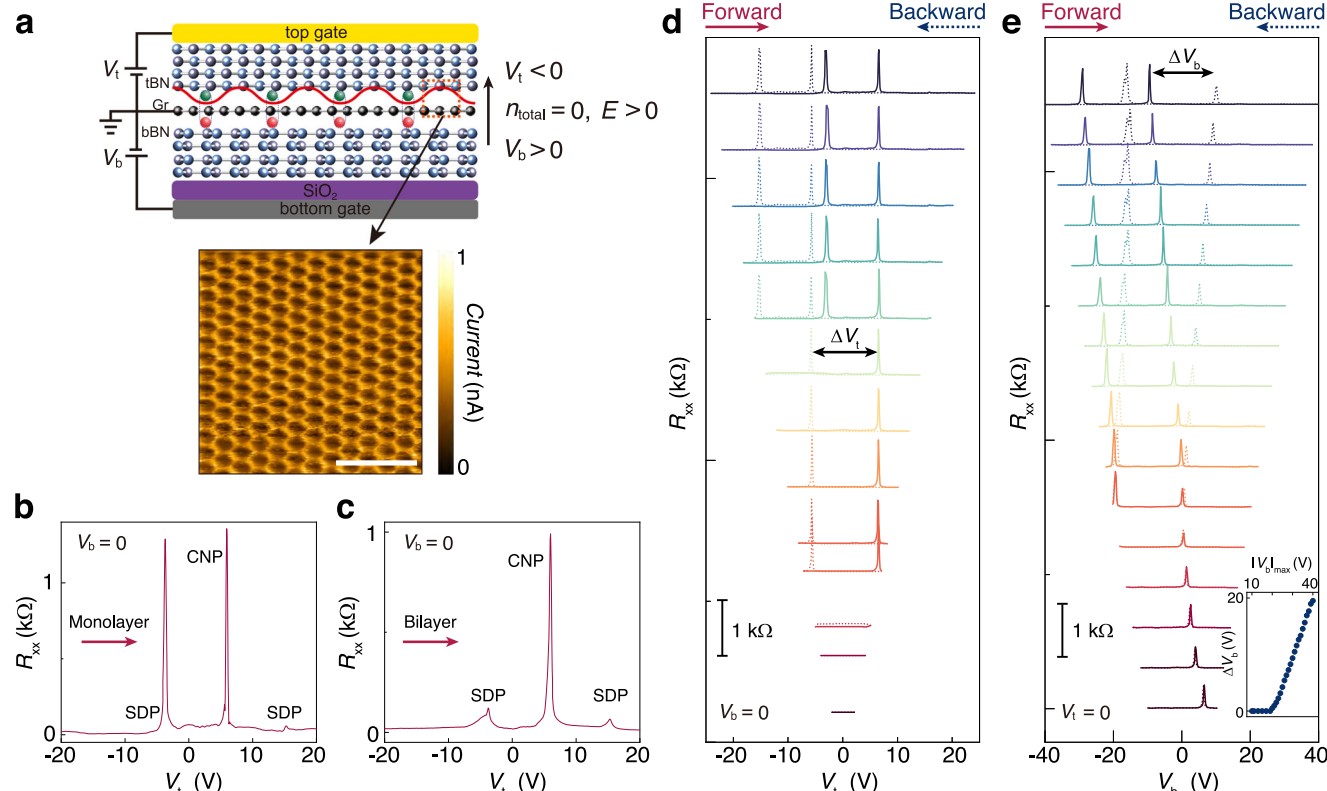

**Fig. 1 | Ferroelectric hysteresis in monolayer graphene superlattices.**
**a** Schematic of our device with asymmetric moiré interfaces and dual-gate structure. The red wavy line illustrates the moiré potential at top interface trapping holes (green balls) injected by the top gate ($V_t$), which are bound by itinerant electrons (red balls) injected by the bottom gate ($V_b$). The vertical arrow defines the positive electric field. Bottom panel shows conductive atomic force microscopy image of graphene/h-BN moiré superlattices. The scale bar is 50 nm. **b, c** Four-terminal longitudinal resistance $R_{xx}$ as a function of $V_t$ at fixed $V_b = 0$ for monolayer

graphene (**b**) and bilayer graphene (**c**) moiré superlattices. The arrows illustrate the sweep direction. **d** $R_{xx}$ as a function of $V_t$ by sweeping $V_t$ in various ranges ($|V_t|_{max}$) while fixing $V_b = 0$. **e** $R_{xx}$ as a function of $V_b$ by sweeping $V_b$ in various ranges ($|V_b|_{max}$) while fixing $V_t = 0$. The curves in (**d**) and (**e**) are vertically shifted for clarity. The forward and backward sweeps are shown in solid and dashed lines, respectively. The inset in (**e**) plots the difference of charge-neutrality points between forward and backward sweeps as a function of $|V_b|_{max}$.

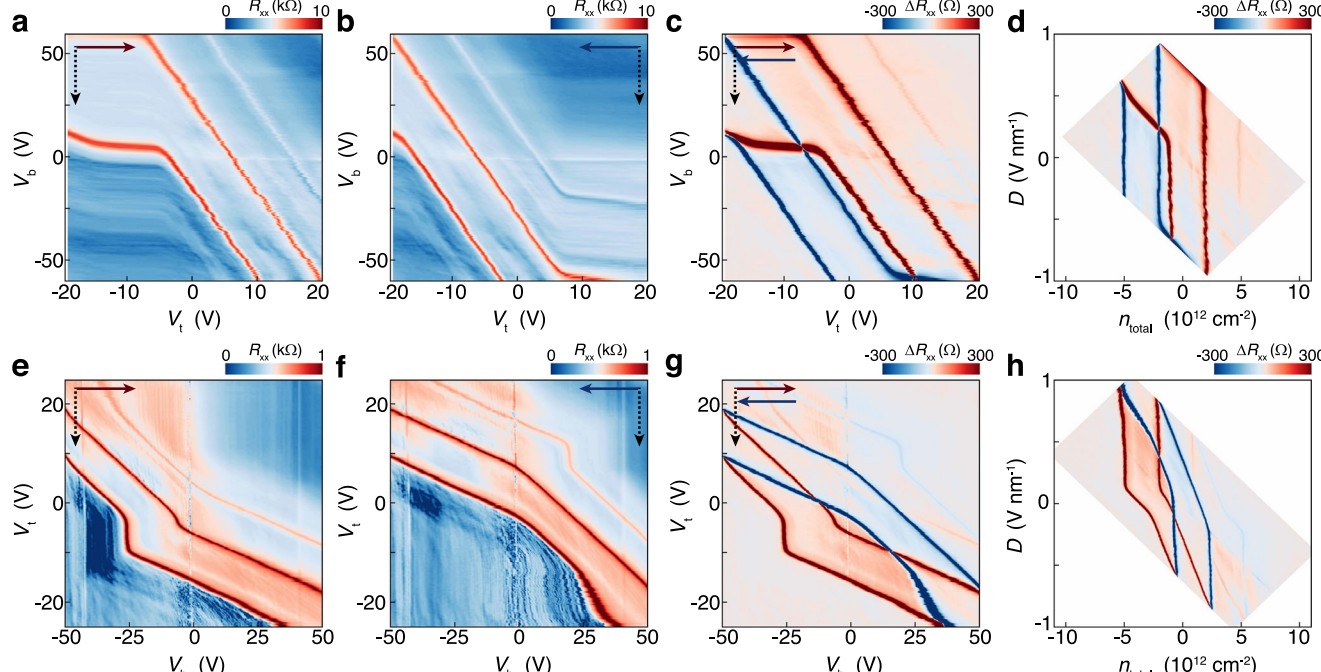

**Fig. 2 | Gate-specific anomalous screening and scan-dependent hysteresis.** Dual-gate maps of $R_{xx}$ by scanning $V_t$ forward (**a**) and backward (**b**) at each fixed $V_b$. **c** The difference between $R_{xx}$ in (**a**) and (**b**). **d** The corresponding $n_{total} - D$ plot of (**c**). Dual-gate maps of $R_{xx}$ by scanning $V_b$ forward (**e**) and backward (**f**) at each fixed $V_t$. **g** The difference between $R_{xx}$ in (**e**) and (**f**). **h** The corresponding $n_{total} - D$ plot of (**g**). In (**a**–**c**, **e**–**g**), the fast-scan axis and slow-scan axis are plotted in horizontal axis and vertical axis, respectively. The solid arrows illustrate the fast-scan direction, and the dashed arrows mark the slow-scan direction.

moiré superlattices hallmark with two satellite peaks at second Dirac point (SDP) besides the main Dirac peak at charge-neutrality point (CNP). The twist angle between graphene and top h-BN is calculated to be 0.68°, resulting in a moiré wavelength of about 12.4 nm (see Supplementary Note 2).

To unveil the layer-dependent ferroelectricity in graphene moiré superlattices, we fabricated a device comprising regions of monolayer, bilayer, and trilayer graphene, which allows us to in-situ compare their transport behaviors. The layer number of each region can be easily distinguished from the optical contrast and further confirmed by Raman spectra (see Supplementary Fig. 1). Figure 1b, c show the transport behaviors measured from monolayer and bilayer graphene moiré superlattices, respectively. Compared with bilayer graphene moiré superlattice, the monolayer counterpart exhibits prominent electron-hole asymmetry with hole-side SDP reaching the same order as CNP and weak electron-side SDP, which is consistent with those reported in literatures[31–35]. More characteristics of monolayer graphene can be identified from the nearly $D$-independent CNP (see Supplementary Fig. 2). In the main text, we mainly present the data from monolayer graphene device (Device D1-1), leaving the data from other devices including bilayer, trilayer and twisted bilayer graphene in Supplementary Information. All the data were taken at the base temperature $T = 2.2$ K unless otherwise specified.

Figure 1d, e show the transfer characteristics of monolayer graphene device given by sweeping either the top gate ($V_t$) or bottom gate ($V_b$) forward and backward, while keeping the other gate at zero. For $V_b$ sweeps, the forward and backward curves overlap within small gate-sweep ranges ($|V_b|_{max} \leq 19$ V). However, when $|V_b|_{max} > 19$ V, four-terminal longitudinal resistance $R_{xx}$ exhibits remarkable hysteresis, which can be easily identified by tracking the positions of CNP and SDP. We can exclude the extrinsic origins of this hysteresis (see Supplementary Note 3). Although both $V_t$ and $V_b$ sweeps show hysteresis, they exhibit different responses across various sweeping ranges. Exceeding a critical $|V_b|_{max}$, the amplitudes of hysteretic loops measured by the shift of CNP ($\Delta V_b$) increase with the increasing $|V_b|_{max}$ as

shown in the inset of Fig. 1e, while $\Delta V_t$ are almost unchanged in $V_t$ sweep as shown in Fig. 1d. The distinct gate tunability reflects the asymmetric moiré potential at the two interfaces of graphene.

## Gate-specific anomalous screening

It's worth noting that in Fig. 1d, hole-side SDP does not appear as expected when $|V_t|_{max} \leq 14$ V even if the sweep range exceeds the full filling of moiré band (corresponding $\Delta V_t' = 9.7$ V in this device). To further investigate this anomaly, we measured the dual-gate maps of $R_{xx}$ shown in Fig. 2, which strongly depend on the scan directions of both top and bottom gates. For example, when comparing the $V_t$ forward (Fig. 2a) with backward (Fig. 2b) maps, remarkable hysteresis can be observed by tracing the positions of CNP and SDPs. The subtraction between the forward and backward maps is plotted in Fig. 2c. Three sets of hysteretic loops consisting of red and blue peaks are associated with CNP and two SDPs.

Besides the hysteresis behavior, another prominent feature in Fig. 2a is the nearly horizontal lines, manifesting as $R_{xx}$ anomalously independent of $V_t$. In normal dual-gate maps of graphene devices, the resistance peaks associated with CNP and SDPs trace as straight diagonal lines whose slopes are determined by the capacitances of two gate dielectrics, suggesting that both gates can effectively inject carriers and tune the Fermi energy. If the dual-gate maps are converted to $n_{total}$ - $D$ maps, the diagonal lines turn into vertical lines (see Methods). However, in our device, within some regions (for instance, $-20$ V $< V_t < -5$ V for hole-side SDP in Fig. 2a), the position of SDP abruptly freezes, as if $V_t$ has been screened and does not work anymore. We denote this phenomenon as GSAS. In this device, $V_t$ is the specific gate, stemming from moiré superlattice at the top interface (see Supplementary Note 2). The appearances of GSAS regions depend on the scan direction of fast-scan gate as shown in Fig. 2a, b. As the fast-scan gate is changed from $V_t$ to $V_b$ as shown in Fig. 2e, f, the GSAS regions tend to shrink but still exist. Moreover, the slopes of the diagonal line are different before and after GSAS occurs. For instance, if tracing the electron-side SDP in Fig. 2f, the slope of diagonal line is

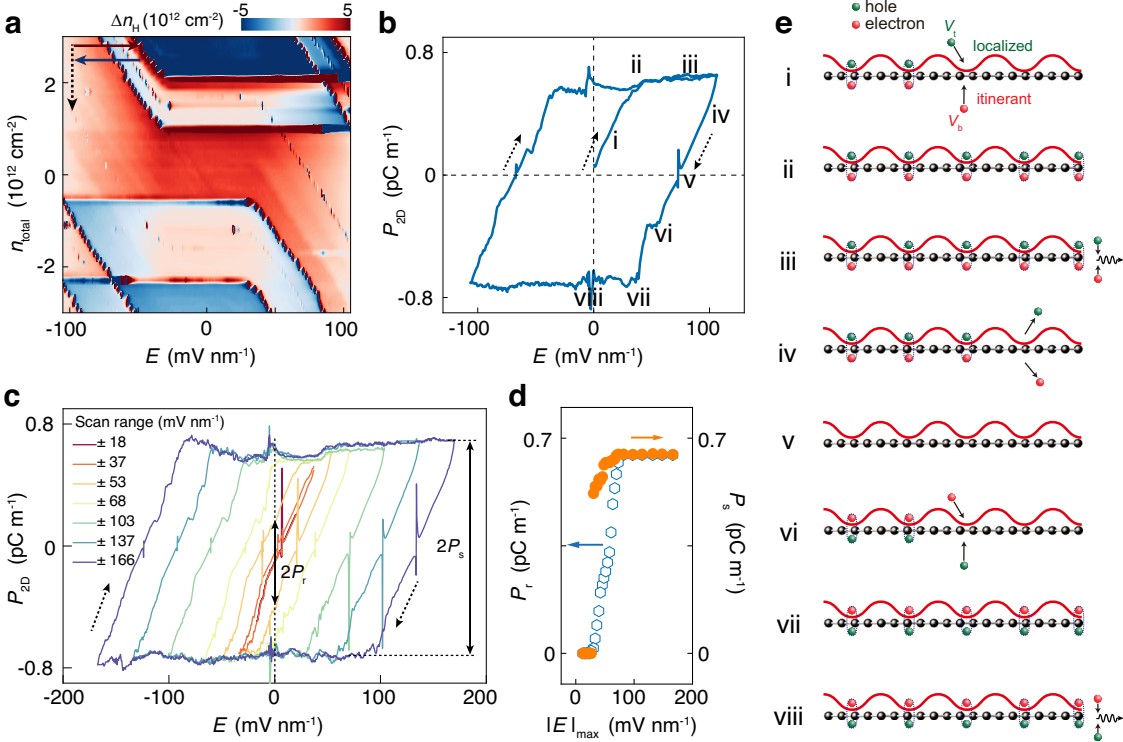

**Fig. 3 | Polarization-electric field ($P_{2D}$-$E$) hysteresis loops. a** The difference of Hall carrier density $n_H$ between the forward and backward sweeps of external electric field $E$ at each fixed carrier density $n_{total}$. **b** Two-dimensional polarization $P_{2D}$ as a function of $E$ measured by sweeping $E$ sequentially in the direction denoted by the arrows at fixed $n_{total} = 0$. **c** Scan-range dependent $P_{2D} - E$ hysteresis loops measured by the same method as that in (**b**). The remanent polarization $P_r$ and saturation polarization $P_s$ are extracted according to the marks in the plots. **d** Summary of $P_r$ and $P_s$ as a function of scan range $|E|_{max}$. **e** Schematic of charge polarization and saturation at each process marked in (**b**). The red wavy lines illustrate the moiré potential. The green (red) ball denotes hole (electron). The arrows illustrate the injection (or extraction) process of charge carriers at top and bottom interfaces controlled by $V_t$ and $V_b$, respectively. The black wavy lines in Process iii and viii denote the recombination of electrons and holes. Process i, iv, v, and vi are GSAS regions. Process ii, iii, vii, and viii are the normal dual-gating regions. All the data in (**b**) and (**c**) were measured at fixed $n_{total} = 0$. The sharp peaks near $P_{2D} = 0$ in (**b**, **c**) are due to the measured $R_{xy} \to 0$ when passing through CNP.

normal when the slow-scan gate ($V_t$) is swept from 25 to 12 V. Across the region of 12 V>$V_t$>7 V, electron-side SDP appears as a vertical line indicating $V_t$ is screened. When $V_t$<7 V, trajectory of electron-side SDP becomes a diagonal line but with a smaller slope compared with initial states ($V_t$>12 V), indicating that $V_b$ is partially screened in this region. The CNP and hole-side SDP show similar evolution trend, although occur at slightly different regions. The GSAS can be identified more clearly as we plot the corresponding $n_{total}$ - $D$ maps in Fig. 2d, h, where straight lines represent the normal gating effect and oblique lines indicate screening effect. More detailed results show that GSAS also depends on the scan range of fast-scan gate (see Supplementary Figs. 9 and 10), but independent of the scan direction of slow-scan gate (see Supplementary Fig. 11).

**Polarization-electric field hysteresis loops**

The presence of GSAS indicates the anomalous external field tunability of charge carrier density in this system. To unveil this, we resort to the Hall measurements. The Hall resistance $R_{xy}$ directly probes the itinerant charge density given by $n_H = -B/eR_{xy}$, where $B$ is external magnetic field, and $e$ is the electron charge. Figure 3a shows the difference of Hall density ($\Delta n_H$) between forward and backward sweeps as a function of $n_{total}$ and $E$, in which the fast-scan axis is $E$, achieved by simultaneously sweeping $V_t$ and $V_b$ with a fixed relation (see Methods). In normal dual-gate devices, scanning $E$ will not change $n_H$ (=$n_{total}$) when fixing $n_{total}$. Specifically, at $n_{total} = 0$, we should achieve $n_H = 0$. However, in our device, as shown in Fig. 3b,c, though $n_{total}$ is fixed at zero, prominent nonzero polarization $P_{2D} = en_H d_{dipole} \propto n_H$ (see Methods) can be achieved and is strongly dependent on the scan history of $E$. Similar behavior of tunable $\Delta n_H$ by scanning $E$ at $n_{total} \neq 0$

can be observed as shown in Fig. 3a. We believe this feature shares common origin as GSAS, which we will discuss later.

Figure 3b shows a typical $P_{2D} - E$ hysteresis loop. Obviously, two distinct states of nonzero remanent polarization ($P_r$) with opposite signs can be achieved at $E = 0$ after a history of applying $E$. The signs of the two states can be reversed by applying a large $E$ with the opposite direction. More generally, the polarization at finite $E$ is dependent not only on the current $E$ but also on its applied history, yielding a hysteretic loop. From Fig. 3c, d, we find no obvious hysteresis within the sweep range $|E|_{max}$<28 mV nm⁻¹. This feature is a typical ferroelectric characteristic that switching polarization necessitates $E$ exceeding a critical value. With increasing the sweep range of $|E|_{max}$>28 mV nm⁻¹, the hysteresis loops and the corresponding $P_r$ dramatically increase, resulting in a large memory window with two nonvolatile states. All the above features, including both spontaneous polarization and its switchable characteristic, unambiguously confirm the observations of ferroelectricity in our system. When $|E|_{max}$ is increased to ~ 73 mV nm⁻¹, $P_r$ approaches the saturation polarization ($P_s$). Further increasing $|E|_{max}$ does not change $P_r$ anymore. However, the window of hysteresis loops continuously enlarges without showing any sign of saturation with increasing $|E|_{max}$. Figure 3d summarizes the $P_r$ and $P_s$ as a function of $|E|_{max}$, showing a step-like increase in $P_r$ and $P_s$.

**Mechanism of electronic ferroelectricity**

Likewise, the measurements of $P_{2D} - E$ hysteresis loops provide us essential information to understand the mechanism of the observed ferroelectricity in our system. Herein, the hysteresis loops are divided into eight processes in a half cycle as shown in Fig. 3e. The other half cycle has similar processes. Normally, when we fix total carrier density

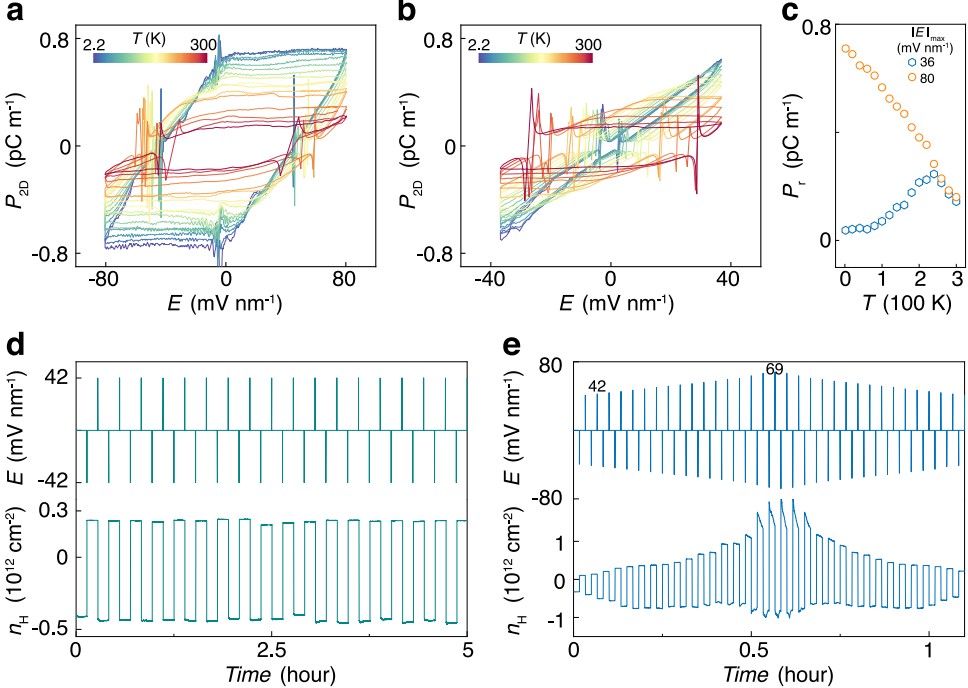

**Fig. 4 | Temperature-dependent ferroelectricity and nonvolatile switching.** $P_{2D} - E$ hysteresis loops measured at various temperature for two representatives $|E|_{max} = 80$ mV nm$^{-1}$ (**a**) and 36 mV nm$^{-1}$ (**b**). The traces in (**a**, **b**) are measured from $T = 2.2$ K to $T = 300$ K with an interval of 20 K. **c** Temperature dependent $P_r$ at two representatives $|E|_{max} = 36$ mV nm$^{-1}$ and 80 mV nm$^{-1}$. Additionally, we observed $P_s$ decreased with the increasing of temperature for both (**a**, **b**) and the windows of $P_{2D} - E$ hysteresis loops were broadened at elevated temperature. **d** $n_H$ measured as a function of time under repetitive $E$ pulses alternating between two equal values

in amplitude but opposite sign. The upper panel shows repeated cycles of applied $E$. **e** Programmable measurement of $n_H$ in response to a series of $E$ pulses. $E$ pulses were programmed to be alternate positive and negative values with gradually increasing amplitudes. The bottom panel is the corresponding $n_H$ as a function of time. All the $n_H$ in (**d**, **e**) are measured from $R_{xy}$ at $B = 0.1$ T without anti-symmetric processing. Therefore, there is slightly difference between positive and negative $n_H$ due to the unperfect Hall bar geometry. All the data are measured at fixed $n_{total} = 0$.

$n_{total} = 0$ and scan the electric field $E$, the hole density injected by $V_t$ is equal to the electron density injected by $V_b$. Thus, we will get zero Hall carrier density $n_H = n_{total} = 0$. However, in our devices, the holes injected by $V_t$ (green balls in Fig. 3e) are trapped by top moiré potential, which become localized and don't contribute to channel conductance. This situation is analogous to the case where the holes injected by $V_t$ are zero. Meanwhile, the electrons can be injected by $V_b$ (red balls in Fig. 3e) normally and become detectable by Hall measurement. Therefore, the net carrier density $n_H$ is nonzero.

In Process i shown in Fig. 3b, e, localized holes and itinerant electrons are continuously injected by $V_t$ and $V_b$, respectively. $n_H$ is gradually increased, resulting in a nonzero polarization. The electron-hole dipole moments are formed by the localized holes at top interface and itinerant electrons at bottom interface. It's worth noting that the bound of localized hole and itinerant electron pairs is dynamic, resembling the formation of Cooper pairs in superconductivity. On reaching Process ii, the density of the injected holes by $V_t$ approaches the half-filling of the moiré band. Further increasing $E$ will trigger Process iii. In this process, the saturated moiré band cannot trap the additionally injected holes by $V_t$ anymore. Instead, they recombine with the electrons injected by $V_b$. Therefore, $n_H$ is saturated, and the dual-gating effect becomes valid. In Process iv, $E$ is decreased, namely, holes and electrons are extracted by $V_t$ and $V_b$, respectively. The remaining itinerant electrons (i.e., $n_H$) continuously decrease till zero, arriving at Process v. Further decreasing $E$ will inject localized electrons at top interface and itinerant holes at the bottom interface, as shown in Process vi. The direction of electron-hole dipole moment (i.e., polarization) is reversed, relative to Process i-iv. In the subsequent Process vii-viii, $n_H$ saturates again, similar to Process ii-iii but with opposite values.

Besides the standard hysteresis loop, we also observed several unique switching behaviors which can support the mechanism aforementioned.

Firstly, conventional ferroelectric insulators always show an anticlockwise hysteretic $P_{2D} - E$ loop because the bound charges only occur at the interface[17,20]. However, ferroelectric semiconductors or metals, due to the presence of mobile carriers, can exhibit partial polarization switching, resulting in clockwise hysteresis[36]. In our system, the direction of hysteretic loops progresses in a clockwise manner as shown in Fig. 3b and c, due to itinerant charges in graphene effectively reducing the penetration depth of electric fields. This feature indicates the electric dipoles and itinerant carriers are in the same channel in our system.

Secondly, the measured saturation itinerant density $n_H^s \approx 1.5 \times 10^{12}$ cm$^{-2}$ is independent of $|E|_{max}$ and layer number (see Supplementary Fig. 16), but highly relative to moiré period. In our device, the observation of SDP facilitates the determination of the full-filling carrier density ($n_{Full} = 3.0 \times 10^{12}$ cm$^{-2}$) of a moiré band. We found that the saturation occurs at half-filling of the moiré band, as we have $n_H^s \approx 0.5 n_{Full}$. Increasing temperature will thermally active the localized carriers, causing them to become itinerant ones. Therefore, we observed the decrease in $n_H^s$ (or $P_s$) with increasing temperature, as shown in Fig. 4. The temperature-dependent $n_H^s$ follows the thermal activation fitting as shown in Supplementary Fig. 8a.

Thirdly, above the threshold ($|E|_{max} > 73$ mV nm$^{-1}$), as shown in Fig. 3b, c the $P_r$ is approximately identical to the $P_s$, indicating the existence of single domain in our sample. The uniform polarization and smooth switching in our system suggest that the ferroelectricity arises more likely from electronic dynamics, rather than sliding ferroelectricity assisted by domain motions. The slope of the curve

tracing from Process iv-v-vi-vii remains unchanged, regardless of variations in $|E|_{max}$ (see Fig. 3c), temperature (see Fig. 4a, b), or layer number (see Supplementary Fig. 16), as it's only relative to the gating capacity of $V_b$. The polarization switching is not due to the domain motion as that in conventional ferroelectricity or sliding ferroelectricity, but arising from the process of injecting and exctracting localized carriers assisted by GSAS as shown in Process iv-v-vi-vii.

Fourthly, when $|E|_{max} < 73$ mV nm$^{-1}$, Process ii is absent, because of the insufficient supply of localized carriers in Process i, resulting in $n_H^r < n_H^s$. This is also the origin of the nonmonotonic dependence of $P_r$ on temperature for $|E|_{max} = 36$ mV nm$^{-1}$ as shown in Fig. 4b, c, because it needs to keep the slope of the GSAS curves unchanged while decreasing $n_H^s$ with increasing temperature. Within the region of $n_H^r < n_H^s$, quasi-continuous remanent polarizations, which are stable even after $E$ is removed, endows the device with the functionality of multi-state data storage and multifunctional synapse emulation.

## Layer independence of electronic ferroelectricity

The GSAS and ferroelectricity observed in monolayer graphene moiré superlattice resemble that in bilayer counterpart[16,26]. Based on previous understanding, layer-polarized flat moiré bands and interlayer charge transfer are essential roles in the emergence of ferroelectricity in graphene moiré systems. To explicitly unveil layer-dependent ferroelectricity, we intentionally designed a control device, which allows us to in-situ compare the emerging ferroelectricity in monolayer, bilayer, and trilayer graphene moiré structures. It's found that all of these three systems exhibit ferroelectric behavior and have some characteristics in common (see Supplementary Figs. 16–19). Firstly, the ferroelectric hysteresis coexists with GSAS. Secondly, the GSAS depends on the gate sweep range and is believed to arise from the asymmetric moiré superlattice. Thirdly, the window of hysteresis loops continuously enlarges without saturation with the increasing $|E|_{max}$ as shown in Fig. 3c and Supplementary Fig. 16b, which is an atypical behavior distinct from conventional ferroelectricity. Above a critical $|E|_{max}$, they have the same saturation Hall carrier density $n_H^s$, which is independent of layer number, but highly relative to half-filling density of a moiré band (see Supplementary Fig. 16). Our results suggest the electronic ferroelectricity observed here is independent of graphene layer number and the fine band structure of graphene (see in Supplementary Figs. 16–19). However, we believe the semimetal characteristics of graphene plays a crucial role since the formation of electron-hole dipoles hinges on the facile excitation of electron-hole pairs by gates.

## Robust ferroelectricity for nonvolatile memory devices

We further study the temperature dependence of electronic ferroelectricity and demonstrate proof-of-concept devices, providing promising applications of monolayer graphene ferroelectricity in multi-state data storage. As shown in Fig. 4a, c, $P_r$ decreases with increasing temperature at $|E|_{max} = 80$ mV nm$^{-1}$, which is opposite to other extrinsic mechanisms such as charge trapping and external adsorbates (see more discussions in Supplementary Note 3), but consistent with the typical ferroelectric behavior. Specially, $P_r$ exhibits nonmonotonic dependence on temperature at $|E|_{max} = 36$ mV nm$^{-1}$ shown in Fig. 4b, c as a unique feature of electronic ferroelectricity. Moreover, we found that the ferroelectric hysteresis loops and spontaneous polarization can persist even at room temperature as shown in Fig. 4a–c. The robustness of our electronic ferroelectricity can be further demonstrated by switching the polarization states using a small $E$ pulse in a nonvolatile way as shown in Fig. 4d. We also check the stability of the nonvolatile switch by keeping the polarization states for an extended period, which remains almost the same for more than 12 h (see Supplementary Fig. 20).

Apart from two-state switch resembling that in conventional ferroelectric materials, the multiple spontaneous polarization states can be achieved here. It's noted that the multi-state switch in our system is

not arising from domain configurations as that in conventional ferroelectric memristors[37]. Instead, the tunable $P_r$ is achieved by the continuous injection of localized holes (electrons) by the specific gate ($V_t$, in this specific device) and electrons (holes) by the other gate. This unique feature endows us to utilize it to realize unconventional applications. As shown in Fig. 4e, positive and negative pulses of $E$ are sequentially applied with gradual increase in the amplitudes. Under pulses of small $|E| < 60$ mV nm$^{-1}$, $|n_H|$ increases quasi-continuously with increasing $|E|$. Each polarization state is stable over time even after $E$ is removed and switchable by changing sign of $E$ pulse. This typical nonvolatile memory state can be widely applied to data storage. Beyond the simple two-state (0 and 1 in digital circuits) storage, our device can function as multiple-state storage.

## Synapse emulation

Our ferroelectric devices exhibit diverse electrical characteristics, enabling us to emulate multifunctional synaptic activities such as synaptic plasticity including short-term plasticity (STP), long-term potentiation (LTP), and long-term depression (LTD). By programming the magnitude and sign of $E$ pulses as shown in Fig. 5a, the nonvolatile and quasi-continuous change of $|n_H|$ can emulate the synaptic plasticity, which is the key component of artificial synaptic devices for neuromorphic computing. In neuroscience, the long-term synaptic plasticity represents the ability of synapses to strengthen (LTP) or weaken (LTD) over time in response to increases or decreases in their

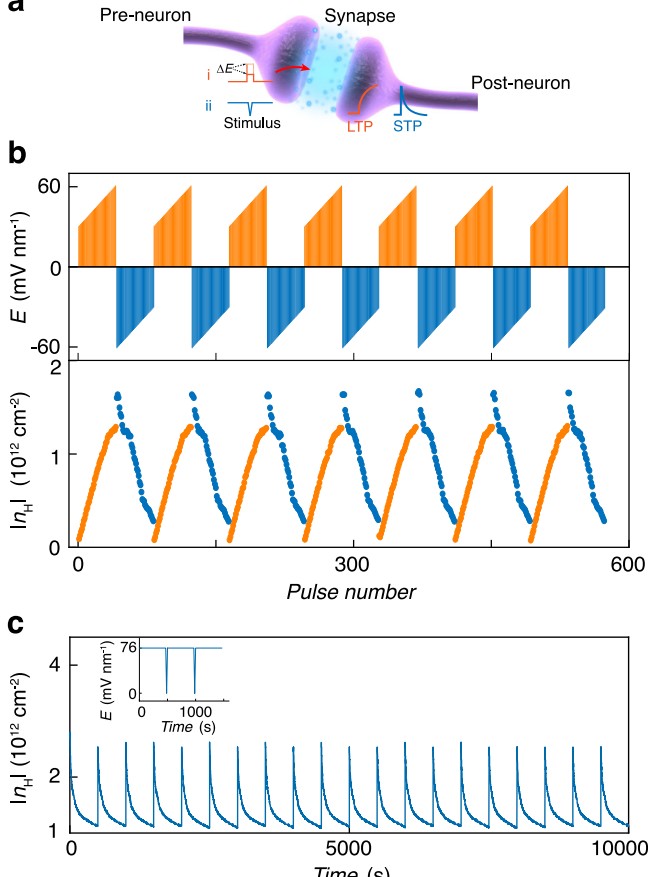

**Fig. 5 | Multifunctional synaptic devices. a** Schematic of a biological synapse. **b** The emulation of long-term plasticity in synaptic devices by utilizing nonvolatile multiple states in our device. $|n_H|$ serves as the synaptic weights, which can continuously increase and decrease under a series of input stimuli (here is $E$), emulating the long-term potentiation and long-term depression, respectively. **c** The emulation of short-term plasticity. $|n_H|$ evolves as a function of time under repetitive stimuli of $E$. The inset shows one cycle of applied $E$.

activity and is widely considered as a primary mechanism for learning and memory. Analogously, in our system, by writing $E$ pulses, the device shows potentiation and depression of $|n_H|$ (emulating synaptic weight), which is reminiscent of LTP and LTD in synapse transistors, respectively.

More excitingly, we notice that $|n_H|$ experiences spontaneous decay in a short time at large $E > 60$ mV nm$^{-1}$, as shown in Fig. 5b. To better understand the behavior under large electric field, we investigated how $|n_H|$ emulating the synaptic weight evolves upon application of a train of $E$ pulses. Upon excitation of a high $E$ pulse, $|n_H|$ immediately increases to a high value and gradually relaxes until the application of next pulse as shown in Fig. 5c. This feature can be used to emulate STP for processing temporal information. This process is highly reproducible and can work for over 20 cycles without any degradation in performance as demonstrated in Fig. 5c. The underlying mechanism is still to be understood but may be attributed to the saturation of moiré traps.

## Discussion

Our findings of electronic ferroelectricity in monolayer graphene not only enrich its fruitful properties in this wonder material but also offer new opportunities for exploring novel physics. When interplaying with other properties in graphene such as ferromagnetism, topology, and superconductivity, intriguing properties including unconventional multiferroics and topological ferroelectrics may emerge. From the perspective of applications, nonvolatile memory and synaptic devices based on monolayer graphene possess unique advantages in terms of high mobility, high stability, and multifunctionality. Moreover, our graphene ferroelectric devices approach to 2D limit, compatible with post-Moore's Law era devices. As aforementioned, the main ferroelectric features in monolayer graphene moiré superlattices can survive up to room temperature, further facilitating their applications. The already well-developed growth of scalable monolayer graphene superlattices can be used to construct the novel ferroelectric devices[38]. The h-BN required for the construction of moiré superlattices also serves as the dielectric material, which simplifies the design of ferroelectric devices.

## Methods

### Device fabrication

All the devices were made using standard dry-transfer method. In brief, graphene and h-BN flakes were mechanically exfoliated on the 285 nm SiO$_2$/Si substrates. In particular, the graphene flake of Device D1 contains consecutive mono-/bi-/tri-layer parts, which were identified by their optical contrast (Supplementary Fig. 1). The top h-BN, graphene, and bottom h-BN flakes were layer-by-layer assembled using a poly(bisphenol A carbonate)/polydimethylsiloxane (PC/PDMS) stamp. Graphene flake was intentionally aligned with both top and bottom h-BN by utilizing their straight and long edges. The alignment between graphene and h-BN can be determined by transport measurement, while the relative angle between top and bottom h-BN was determined by second harmonic generation (SHG) measurements. The h-BN/graphene/h-BN heterostructure was finally released on a plasma-cleaned SiO$_2$/Si substrate which served as back gates. The contact regions were patterned by electron beam lithography (EBL) and etched as trenches by CHF$_3$/O$_2$ plasma. Metallic contacts (5 nm Cr/60 nm Au) were deposited into the trenches. The metallic top gate was made by a second EBL and e-beam evaporation. The final Hall bar geometry was defined by another round of EBL and plasma etching. The list of devices, including the ferroelectricity behaviors and the stagger angles between top and bottom h-BN, are summarized in Supplementary Table 1.

### Optical measurements

Raman measurements were performed at room temperature using a confocal Raman spectrometer (Witec Alpha 300RAS with UHTS 300 spectrometer) equipped with a 532 nm excitation laser. The laser spot size was about 1 μm. The second harmonic generation (SHG) measurements were conducted in the same setup with incident light wavelength of 1064 nm and a fixed excitation power of 20 mW.

### Conductive atomic force microscopy measurements

The sample for conductive atomic force microscopy (c-AFM) measurements was made by sequentially picking up h-BN and graphene with PC/PDMS stamp. The stack was then flipped on a fresh PDMS. After dissolving PC with n-methyl-2-pyrrolidone (NMP), the stack was released on a new SiO$_2$/Si substrate. The c-AFM measurements were performed on Asylum Research Jupiter XR at room temperature.

### Transport measurements

The devices were wire-bonded on the LCC-44 chip carriers (Kyocera) and measured in a 4 K closed cycle refrigerator system (Janis SHI-4-2) cooled down by a Sumitomo F-20L cold head. The base temperature was 2.2 K, recorded by a calibrated sensor mounted near the sample and controlled by a Lake Shore temperature controller. All the measurements presented in the manuscript, such as dual gate mappings and $P_{2D} - E$ loops, except the temperature-dependent measurements, were carried out at this base temperature. The longitudinal and Hall resistances were measured by lock-in amplifier (SR-830) with constant excitation current of 100 nA. The top and bottom gates were applied by source meters (Keithley 2450).

Our dual-gate structure allows us to convert $V_t - V_b$ maps to $n_{total} - D$ maps. The total gate-induced carrier density is $n_{total} = (C_b V_b + C_t V_t)/e$, and the displacement field is $D = (C_b V_b - C_t V_t)/2\varepsilon_0$, where $C_t$ and $C_b$ are the capacitance per area of top and bottom gate measured by Hall effect, respectively, $e$ is the elementary charge, and $\varepsilon_0$ is the vacuum permittivity. The four-terminal longitudinal resistance $R_{xx}$ maps were acquired at zero magnetic fields unless otherwise specified.

To convert $D$ to $E$, we note that $V_b$ was applied on SiO$_2$ and bottom h-BN. Therefore, the voltage across the bottom h-BN is $V_{bBN} = \frac{V_b \varepsilon_{SiO_2} d_{bBN}}{\varepsilon_{BN} d_{SiO_2} + \varepsilon_{SiO_2} d_{bBN}}$, where $d_{bBN}$ and $d_{SiO_2}$ are the thickness of bottom h-BN and SiO$_2$, respectively[39], $\varepsilon_{SiO_2} = 4.3$ and $\varepsilon_{BN} = 3.8$ are the dielectric constant of h-BN and SiO$_2$, respectively, calibrated by capacitance measurements through Hall effect at normal gating region. The electric field across graphene is $E = (V_{bBN}/d_{bBN} - V_t/d_{tBN})/2 = D/\varepsilon_{BN}$, where $d_{tBN}$ is the thickness of top h-BN.

### $n_H$ and $P_{2D} - E$ loops measurements

The itinerant carrier density $n_H$ is determined from Hall effect by measuring Hall resistance $R_{xy}$ under small magnetic field of $B = \pm 0.1$ T. $R_{xy}$ is anti-symmetrized by $R_{xy} = (R_{xy}^{0.1T} - R_{xy}^{-0.1T})/2$ to remove residual values induced by unperfect sample geometry. The Hall (itinerant) carrier density is calculated to be $n_H = -B/eR_{xy}$. To measure Fig. 3a, we simultaneously swept $V_b$ and $V_t$ with a specific relation by fixing $n_{total} = (C_b V_b + C_t V_t)/e$ as a constant value, and measured $n_H$. In this way, we can realize the sweeps of $E$ at fixed $n_{total}$, namely, $E$ as fast-scan axis and $n_{total}$ as slow-scan axis. The $\Delta n_H$ is the difference of $n_H$ between forward and backward sweeps of $E$.

In order to measure $P_{2D} - E$ loops, we always fixed $n_{total} = 0$. By sweeping $E$ forward and backward in a given rang of $|E|_{max}$, we acquired the response of $n_H$. The 2D polarization $P_{2D}$ follows $P_{2D} = e n_H d_{dipole}$[17,20], where $d_{dipole}$ is the size of dipole moment, i.e., the thickness of graphene. For monolayer graphene, $d_{dipole} = 0.26$ nm[40], while for bilayer graphene, $d_{dipole}$ is doubled as that of monolayer graphene. To compare with other ferroelectrics, we can convert $P_{2D}$ to $P_{3D} = \frac{P_{2D}}{d_{dipole}} = e n_H$. At $|E|_{max} = 80$ mV/nm, the measured saturation itinerant density is $n_H^s \approx 1.5 \times 10^{12}$ cm$^{-2}$, yields 2D saturation polarization $P_{2D} = 0.63$ pC/m, which is corresponding to $P_{3D} = 0.24$ μC/cm$^2$.

This value is in the same order as those in other 2D ferroelectrics, such as bilayer graphene superlattices[16,25], twisted h-BN[17], 3R-MoS$_2$[20], while much smaller than those in other traditional ferroelectrics probably because of its 2D nature.

## Reporting summary

Further information on research design is available in the Nature Portfolio Reporting Summary linked to this article.

## Data availability

Relevant data supporting the findings of this study are available from the corresponding authors upon request. Source data are provided with this paper.

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

## Acknowledgements

This work was funded by National Natural Science Foundation of China (Grant No. 12274354), the Zhejiang Provincial Natural Science Foundation of China (Grant No. LR24A040003; XHD23A2001), and Westlake Education Foundation at Westlake University. We thank Chao Zhang and Zhen Yang from the Instrumentation and Service Center for Physical Sciences (ISCPS) at Westlake University for technical support in data acquisition. We also thank Westlake Center for Micro/Nano Fabrication and the Instrumentation and Service Centers for Molecular Science for facility support. L.Z. acknowledges to the Zhejiang Province Selected Funding for Postdoctoral Research Projects (ZJ2023077). K.W. and T.T. acknowledge support from the JSPS KAKENHI (Grant Numbers 21H05233 and 23H02052) and World Premier International Research Center Initiative (WPI), MEXT, Japan.

## Author contributions

S.X. and N.X. conceived the idea and supervised the project. N.X. and L.Z. fabricated the devices. L.Z. performed the transport measurements with the assistance of J.D., W.Z., N.L., and L.W. H.X. performed AFM measurements. L.Z., W.Z., and H.X. performed Raman and SHG measurements. K.W. and T.T. grew h-BN crystals. L.Z., N.X., and S.X. analyzed the data and wrote the paper. All the authors contributed to the discussions.

## Competing interests

The authors declare no competing interests.

## Additional information

Na Xin or Shuigang Xu.

**Peer review information** *Nature Communications* thanks Martino
Aldrigo and the other anonymous reviewer(s) for their contribution to the
peer review of this work. A peer review file is available.

