## [Transparent Peer Review file · Nature Communications]

Electronic ferroelectricity in monolayer graphene moiré superlattices

Corresponding Author: Professor Shuigang Xu

Version 0:

Reviewer comments:

Reviewer #1

(Remarks to the Author)

Dear Authors,

Your work about the electronic ferroelectricity in monolayer graphene moiré superlattices is inspiring and presents a rigorous experimental characterization, supporting the initial hypothesis. Moreover, the information provided on the fabrication and characterization are enough to be reproduced.

I have some remarks that could be useful to other readers, if your paper is published, as follows:

- 1) Lines 100-101: you wrote "All the data were taken at the temperature $T = 2.2$ K, unless otherwise specified", please motivate this choice.
- 2) "Polarization-electric field hysteresis loops": the polarization is expressed in pC/m, I guess because it is a 2D material, hence there is no volume associated to the ferroelectric material. Even so, I wonder if it were more correct to consider graphene's thickness and, hence, volume, in order to make some comparisons with "traditional" ferroelectrics, for example the nanoscale ones based on hafnium or similar. The polarization here is in the order of 100's of fC/m, whereas in other ultrathin ferroelectrics it can attain values in the order of 0.1 C/m².
- 3) "Layer independence of electronic ferroelectricity": you measured graphene multilayers up to 3 layers. Did you also consider using more than 3 layers? If not, why?
- 4) "Robust ferroelectricity and memory device": is the device encapsulated? If not, how stable is it under air exposure? Was any performance degradation observed in time?
- 5) "Device fabrication": is the fabrication process able to guarantee the repeatability of the measurements?

Reviewer #2

(Remarks to the Author)

In this study, the authors explore the extension of ferroelectric materials into two-dimensional systems, focusing on monolayer graphene. Traditionally, ferroelectricity requires materials with polar crystalline structures, but monolayer graphene's centrosymmetric hexagonal lattice limits its ferroelectric properties. They demonstrate electronic ferroelectricity in monolayer graphene by creating an asymmetric moiré superlattice at the graphene/h-BN interface, where electric polarization arises from electron-hole dipoles. Measurements of Hall carrier density reveal polarization switching up to room temperature, producing hysteresis loops. Notably, they observe similar ferroelectric characteristics across monolayer, bilayer, and trilayer graphene, indicating that layer polarization is not essential. Additionally, they showcase the potential of these structures for multi-state nonvolatile data storage with high retention. This work advances the understanding of ferroelectricity and highlights graphene's promise for high-speed memory applications.

The authors should give some clarification about some points:

1. How does the moiré potential at the top interface influence the behavior of injected holes or electrons in the system?
2. Why do the localized carriers prevent itinerant electrons or holes injected by the other gate (V_b) from being neutralized, leading to a nonzero Hall carrier density (n_H)?

3. In your study, you examined the temperature dependence of electronic ferroelectricity in monolayer graphene. We observed that the remanent polarization (P_r) decreases with increasing temperature at a maximum electric field of 80 mV nm^{-1} , which aligns with typical ferroelectric behavior. Interestingly, for a maximum electric field of 36 mV nm^{-1} , P_r displayed a nonmonotonic dependence on temperature due to an insufficient supply of localized carriers. Additionally, you found that the ferroelectric hysteresis loops and spontaneous polarization remained stable even at room temperature, demonstrating the robustness of this phenomenon. Please how does the temperature dependence of the remanent polarization (P_r) in monolayer graphene differ from other extrinsic mechanisms like charge trapping?

Here's a brief text followed by a question based on your provided content:

4. Your findings of electronic ferroelectricity in monolayer graphene not only enhance its remarkable properties but also open new avenues for exploring novel physics. The interaction of graphene's ferroelectricity with other characteristics, such as ferromagnetism, topology, and superconductivity, may lead to the emergence of unconventional multiferroics and topological ferroelectrics. From an application standpoint, monolayer graphene offers significant advantages for non-volatile memory and synaptic devices, including high mobility, stability, and multifunctionality. Notably, the ferroelectric features in graphene moiré superlattices can persist up to room temperature, making them suitable for advanced applications. Furthermore, the established methods for growing scalable monolayer graphene superlattices simplify the design of ferroelectric devices, as the required h-BN serves as both a substrate and dielectric material.

In your opinion, what advantages does monolayer graphene offer for non-volatile memory and synaptic devices compared to traditional materials?

5.

In your study, we investigated gate hysteresis in graphene devices, particularly its dependence on extrinsic effects such as charge traps from defects, impurities, or adsorbates. Unlike typical behavior, we observed a robust gate hysteresis that remained consistent across different scan rates and cycles. When fixing V_t at 0 V and varying V_b at scan rates from 150 mV s^{-1} to 400 mV s^{-1} , the forward and backward scan curves displayed nearly identical resistance peak positions and magnitudes. Additionally, tests for endurance showed that repeated forward and backward scans of V_b or V_t for 20 cycles resulted in overlapping curves, indicating excellent cycle endurance. This robust hysteresis was also observed in monolayer, bilayer, and trilayer graphene moiré superlattices, suggesting a common underlying mechanism.

What factors contribute to the observed robustness of gate hysteresis in graphene devices, despite the influence of extrinsic effects like charge traps? and how does the endurance of gate hysteresis across multiple scan cycles reflect on the potential applications of monolayer, bilayer, and trilayer graphene moiré superlattices?

In general, the manuscript has a correct methodological structure. But for my opinion the authors should making a major revision on this manuscript. I hope that authors reconsider these all points, and they clarify some issues.

Reviewer #3

(Remarks to the Author)

In the manuscript "Electric ferroelectricity in monolayer graphene moire superlattices" the authors investigate the possibility to

introduce ferroelectricity in graphene, both single-layer and few layer, by encapsulating it with h-BN. Their results are confirmed

for 5 devices, whose geometry is given in Table S1 of Supplementary Information. The results sound interesting, although they lack

an explanation of the microscopic mechanism leading to the observed phenomena.

From the point of view of symmetry, it is known that the out-of-plane polarisation in a material or a heterostructure can be induced only in the case of a structural asymmetry in the z-direction.

However, in one the devices analysed (D2-1), it is clear that the system possesses the horizontal mirror symmetry, since Θ_{BN} (the relative angle between the top and the bottom h-BN layer) is equal to zero. Thus, it is for me unclear how

in such a configuration the polarisation can arise. It would be beneficial if the authors explain whether the top and bottom h-BN are equal and if yes, give an appropriate explanation of the phenomenon measured.

Version 1:

Reviewer comments:

Reviewer #1

(Remarks to the Author)

Dear Authors,

After evaluating your revised manuscript, I am content with the provided responses to my previous concerns. No more comments from my side.

Reviewer #2

(Remarks to the Author)

The authors are answer to my questions and are improved their manuscript in my opinion this work meriti th spublications in Nature Communications.

Reviewer #3

(Remarks to the Author)

I thank the reader for the answers provided, especially for pointing to the importance of the asymmetric potential.

I would have to disagree with the authors that the device D2-1 does not have the horizontal mirror symmetry. To prove this, we can set the $z=0$ to discuss the position of the graphene monolayer. In this case, the presence of horizontal mirror symmetry implies $f(z)=f(-z)$, meaning that the twist angle between the top hBN and the bottom hBN (θ_{BN}) has to be equal. This is written in Table S1., where Θ_{BN} is defined as the relative angle between the top h-BN and bottom h-BN determined by optical images and SHG measurements.

Additionally, in the suggested reference Nature 588, 71–76 (2020) it is stated that "in the presence of ferroelectric ordering, the interlayer displacement field has two contributions, the field induced by gating and the internal electric field arising from the built-in ferroelectric polarization."

Have the authors distinguished the contribution of the interlayer displacement field due to gating and the built-in ferroelectric polarization? For me it is clear that the device can show polarization due to different top and bottom gating potentials but this does not imply that the sample has built-in ferroelectric polarization, since the asymmetric potential can come solely due to gating.

I cannot recommend the manuscript for publication without adequately addressing these questions.

Version 2:

Reviewer comments:

Reviewer #3

(Remarks to the Author)

I appreciate the authors for their thorough responses and recommend that the manuscript be published in its current form.

Point-by-point response to the Reviewers

We sincerely thank the reviewers for taking the time to assess our manuscript and raising constructive suggestions to improve it. We believe that this letter and the revised manuscript fully addressed your questions and comments. In the following, we respond to the comments in detail. Throughout this letter, we mark light blue color for the comments by the reviewers and black color for our responses. Changes made in the updated manuscript are highlighted in yellow.

Reviewer #1 (Remarks to the Author):

Dear Authors,

Your work about the electronic ferroelectricity in monolayer graphene moiré superlattices is inspiring and presents a rigorous experimental characterization, supporting the initial hypothesis. Moreover, the information provided on the fabrication and characterization are enough to be reproduced.

Our response: We thank you for appreciating the quality and merits of our work. In the following, we address the comments and questions in detail.

I have some remarks that could be useful to other readers, if your paper is published, as follows:

1) Lines 100-101: you wrote "All the data were taken at the temperature $T = 2.2$ K, unless otherwise specified", please motivate this choice.

Our response: We thank you for pointing out this issue. We apologize for the unclear description of our measurements. The transport measurements were performed in a 4 K closed cycle refrigerator system (Janis SHI-4-2) cooled down by a Sumitomo F-20L cold head. The base temperature was 2.2 K, recorded by a calibrated sensor mounted near the sample and controlled by a Lake Shore temperature controller. All the measurements presented in the manuscript, such as dual gate mappings and $P - E$ loops, except the temperature-dependent measurements, were carried out at this base temperature.

In the revised manuscript, we have added the above experimental details in the Method section.

2) "Polarization-electric field hysteresis loops": the polarization is expressed in pC/m, I guess because it is a 2D material, hence there is no volume associated to the ferroelectric material. Even so, I wonder if it were more correct to consider graphene's thickness and, hence, volume, in order to make some comparisons with "traditional" ferroelectrics, for example the nanoscale ones based on hafnium or similar. The polarization here is in the order of 100's of fC/m, whereas in other ultrathin ferroelectrics it can attain values in the order of 0.1 C/m^2 .

Our response: We thank you for your fruitful comments. Indeed, we initially used P_{2D} with a unit of pC/m because our system is a 2D system. As your suggestion, if we consider monolayer

graphene's thickness ($d_{\text{dipole}} = 0.26 \text{ nm}$), we can convert P_{2D} to $P_{3D} = \frac{P_{2D}}{d_{\text{dipole}}} = en_{\text{H}}$, where n_{H} is

the 2D carrier density. As a comparison, the measured saturation itinerant density $n_{\text{H}}^{\text{S}} \approx 1.5 \times 10^{12} \text{ cm}^{-2}$, yields 2D saturation polarization $P_{2D} = 0.63 \text{ pC/m}$, which corresponds to $P_{3D} = 0.24 \text{ } \mu\text{C/cm}^2 = 0.024 \text{ C/m}^2$. This value is in the same order as those in other 2D ferroelectrics, such as bilayer graphene superlattices (*Nature* 588, 71-76, 2020; *Nat. Commun.* 13, 6241, 2022), twisted h-BN (*Science* 372, 1458-1462, 2021), 3R-MoS₂ (*Nat. Nanotech.* 17, 367-371, 2022),

while much smaller than those in other traditional ferroelectrics probably because of its 2D nature.

In the revised manuscript, we have added the above discussions in the Method section.

3) "Layer independence of electronic ferroelectricity": you measured graphene multilayers up to 3 layers. Did you also consider using more than 3 layers? If not, why?

Our response: Thank you for this suggestive comment.

In our work, the asymmetric moiré potential at the interfaces is the crucial factor. Therefore, in principle, the layer number won't affect the electronic ferroelectricity. However, when the layer number exceeds three layers, strong bulk screening effects originating from the large density of itinerant charges will impact the measurements and data analysis. These screening effects may intertwine with the gate-specific anomalous screening effect. Meanwhile, the n_H measured from Hall effect may only reflect the contribution from only one surface and may not directly represent the total polarizations. Therefore, we focus exclusively on the measurements of 1-3 layers. We think our data provide sufficient evidence that layer polarization is not essential for observing the ferroelectricity.

4) "Robust ferroelectricity and memory device": is the device encapsulated? If not, how stable is it under air exposure? Was any performance degradation observed in time?

Our response: We thank you for raising this point.

Yes, all the devices were encapsulated with both top and bottom h-BN. This encapsulated structure has been shown to provide excellent stability, protecting the devices from oxygen and moisture during air exposure (*Science* 342, 614-617, 2013; *Nat. Commun.* 6, 7315, 2015). Our devices exhibit quite stable performance, without observable degradation after multiple times of thermal cycle and sample loading. To further demonstrate the stability of our devices, in the revised manuscript, we re-cooled down the device and re-measured the pulse response, with the time interval of 6 months. The data are as shown in Fig. R1. Instead of Hall carrier density n_H (shown in Supplementary Fig. 20), this time, we directly measured channel current I_{ds} after applying positive and negative pulse for 48 hours, indicating the polarization states remain unchanged. Furthermore, we also repeated the data of scanning V_t forward and backward with increasing range as depicted in Fig. 1d of main text, as shown in Fig. R2. The result is quite reproducible.

In the revised manuscript, we added the above new data to Supplementary Fig. 20 and Fig. 21.

Fig. R1 | Measurement of polarization switching after 6 months since initial measurements.

Fig. R2 | Initial (a) and repeated (b) measurements of R_{xx} versus V_t by sweeping V_t in various ranges ($|V_t|_{\max}$) while fixing $V_b = 0$. The scanning range $|V_t|_{\max}$ increases from 2 V to 12 V with interval of 1 V. The curves in (a) and (b) are vertically shifted for clarity. The forward and backward sweeps are shown in solid and dashed lines, respectively.

5) "Device fabrication": is the fabrication process able to guarantee the repeatability of the measurements?

Our response: Thank you for this comment. We have tried our best to provide the fabrication details as much as possible in the "Device fabrication" section. We have used standard dry-transfer techniques with PC/PDMS stamps to assemble the heterostructure. This h-BN encapsulated structures guarantee the high-performance of graphene devices. The straight edges of graphene and h-BN were intentionally aligned. The alignment can be further checked by SHG

measurements. The fabrication of electrodes was conducted by using standard EBL, plasma etching, and E-beam evaporations with several steps. With this process, we have fabricated several devices, summarized in Supplementary Table 1. Therefore, we believe our process is able to guarantee the repeatability of the measurements.

Reviewer #2 (Remarks to the Author):

In this study, the authors explore the extension of ferroelectric materials into two-dimensional systems, focusing on monolayer graphene. Traditionally, ferroelectricity requires materials with polar crystalline structures, but monolayer graphene's centrosymmetric hexagonal lattice limits its ferroelectric properties. They demonstrate electronic ferroelectricity in monolayer graphene by creating an asymmetric moiré superlattice at the graphene/h-BN interface, where electric polarization arises from electron-hole dipoles. Measurements of Hall carrier density reveal polarization switching up to room temperature, producing hysteresis loops. Notably, they observe similar ferroelectric characteristics across monolayer, bilayer, and trilayer graphene, indicating that layer polarization is not essential. Additionally, they showcase the potential of these structures for multi-state nonvolatile data storage with high retention. This work advances the understanding of ferroelectricity and highlights graphene's promise for high-speed memory applications.

Our response: We thank you for the precise summary of our main findings and recognizing our novelty. We are deeply encouraged that our results are valuable for advancing graphene-based ferroelectric and related applications.

The authors should give some clarification about some points:

1. How does the moiré potential at the top interface influence the behavior of injected holes or electrons in the system?

Our response: We thank you for this question. In a normal graphene/h-BN field effect transistor, the resistance of the channel is effectively modulated by the gate voltage. However, in our device, within some regions (for instance, labeled as double-side arrow between $-15 \text{ V} < V_t < -3 \text{ V}$ for CNP in Fig. R3a), the resistance of CNP abruptly freezes, as if V_t has been screened and does not work anymore. The anomalous screening and hysteresis of top gate is due to that the accumulated charges at graphene/h-BN interface are localized. In other ways, we can view the moiré as a potential well for trapping injected charges. The charge trapping process can be unveiled in Fig. R3b. For instance, we fix $V_b = 41 \text{ V}$ then scan V_t from -15 V to -3 V . V_t firstly extracts localized holes (Process i and ii) and then injects localized electrons till the half filling of a moiré band (Process iii and iv), thus making no modification of the Fermi level E_f . As a result, the corresponding resistance of CNP doesn't change. This process is gate-specific anomalous screening (GSAS), manifesting as horizontal lines as marked in Fig. R3a.

Fig. R3 | **a**, Difference in R_{xx} between forward and backward scan as a function of V_b and V_t . The solid (dashed) arrows illustrate the fast (slow)-scan direction. The scan ranges of V_t is ± 15 V. The width (W) of the GSAS regime corresponds to the screening capability of V_t . **b**, Schematic of charge trapping at top moiré interface marked as double-side arrow in **(a)**. The red wavy lines illustrate the moiré potential. The blue (red) ball denotes hole (electron). The arrows illustrate the injection (or extraction) process of charge carriers by V_t . The right panel illustrates the band structure and Fermi level E_f at each process. E and k represent energy and momentum respectively. Process i to iv are GSAS regions.

2. Why do the localized carriers prevent itinerant electrons or holes injected by the other gate (V_b) from being neutralized, leading to a nonzero Hall carrier density (n_H)?

Our response: Thank you for the question. We apologize for the misleading description. The charges are injected by V_t and V_b independently but exhibit different transport behaviors. Associated with the last response, we would like to take this opportunity to explain the microscopic processes in detail. In Process i of our typical $P_{2D} - E$ hysteresis loops measurement (Fig. R4b and c), we fix total carrier density $n_{total} = 0$ then scan the electric field E from 0 to a certain positive value, that is, the hole density injected by V_t is equal to the electron density injected by V_b . Normally, we will get zero Hall carrier density $n_H = n_{total} = 0$ as expected in Fig. R4a. However, in our devices, the holes injected by V_t (represented by green balls in Fig. R4c) are trapped by top moiré potential, which become localized and don't contribute to channel conductance. This situation is analogous to the case where the holes injected by V_t are zero. Meanwhile, the electrons can be injected by V_b (red balls in Fig. R4c) normally and are detectable by Hall measurement. Therefore, the net carrier density n_H is nonzero.

In the revised manuscript, we have included above discussion in the main text.

Fig. R4 | **a**, Schematic of Hall carrier density n_H measured by scanning electric field E with total carrier density $n_{\text{total}} = 0$ in a normal dual-gated graphene device. **b**, Hall carrier density n_H as a function of E in our monolayer graphene superlattice device measured at $n_{\text{total}} = 0$. **c**, Schematic of charge polarization and saturation at each process marked in **(b)**. Here, we highlight the Process i, in which the holes injected by V_t (green balls) are trapped by top moiré potential, the electrons injected by V_b (red balls) are itinerant.

3. In your study, you examined the temperature dependence of electronic ferroelectricity in monolayer graphene. We observed that the remanent polarization (P_r) decreases with increasing temperature at a maximum electric field of 80 mV nm^{-1} , which aligns with typical ferroelectric behavior. Interestingly, for a maximum electric field of 36 mV nm^{-1} , P_r displayed a nonmonotonic dependence on temperature due to an insufficient supply of localized carriers. Additionally, you found that the ferroelectric hysteresis loops and spontaneous polarization remained stable even at room temperature, demonstrating the robustness of this phenomenon. Please how does the temperature dependence of the remanent polarization (P_r) in monolayer graphene differ from other extrinsic mechanisms like charge trapping?

Our response: We thank you for giving us this opportunity to clarify the mechanism. As illustrated in Fig. R5a, gate hysteresis induced by extrinsic effects should exhibit enhanced performances at high temperature because the charges acquire high energy to migrate. However, this is not the case in our system. We have measured longitudinal R_{xx} by forward and backward sweeping V_t at 2.2 K, 100 K and 200 K while fixing $V_b = 0$ in our monolayer graphene superlattice device, which shows a reverse behavior. Hysteresis tends to be weaker at the higher temperature. Such a behavior can also be manifested by both dual-gate mappings at elevated temperature and thermal-activated remanent polarization P_r at $|E|_{\text{max}} = 80 \text{ mV nm}^{-1}$. In Fig. R6a to Fig. R6c, the GSAS regions (marked by double-headed arrow) shrink gradually with increasing temperature, showing the behavior opposite to charge trapping effects.

The nonmonotonic evolution of P_r with temperature at $|E|_{\text{max}} = 36 \text{ mV nm}^{-1}$ provides additional

evidence for the underlying mechanism described in Fig. 3b and 3e. When $|E|_{\max} < 73 \text{ mV nm}^{-1}$, the insufficient supply of localized carrier in Process i (see in Fig. R4b and R4c) results in $n_{\text{H}}^r < n_{\text{H}}^s$ (or equivalent to $P_r < P_s$) at the base temperature. Meanwhile, one key feature of the $P_{2\text{D}} - E$ hysteresis loops is that the slope (color dashed lines in Fig. R6e) of the parallelogram loops is a constant, independent of temperature, which is determined by the capacitance of the gate dielectric (h-BN). The moiré trapping effect becomes weaker at higher temperature, leading to a smaller saturation polarization P_s value. To keep the slope unchanged, the windows of parallelogram loop need to be larger.

In this scenario, taking $|E|_{\max} = 36 \text{ mV nm}^{-1}$ as an example, when the temperature increases from $T = 2.2 \text{ K}$ to $T \sim 240 \text{ K}$, $P_r (< P_s)$ increases while P_s decreases. At $T \sim 240 \text{ K}$, $P_r = P_s$ is reached. As the temperature further increases from 240 K to 300 K , since the relationship of $P_r = P_s$ is unchanged, P_r decreases accompanied by P_s .

In summary, from the unique temperature-dependent measurements, we can confirm that the hysteresis loops arise from the moiré trapping effects, rather than the extrinsic mechanisms like charge trapping.

In the revised manuscript, we have included Fig. R5 into Supplementary Fig. 6.

Fig. R5 | **a**, Schematic of charge-trapping hysteresis in a graphene device. From left to right, hysteresis becomes more pronounced with increasing temperature. For the actual experiment, please refer to ref [*Phys. Rev. B* 93, 045403 (2016)]. **b**, R_{xx} as a function of V_t at 2.2 K, 100 K and 200 K while fixing $V_b = 0$ in our monolayer graphene superlattice devices.

Fig. R6 | **a**, Difference in R_{xx} between forward and backward scan as a function of V_t and V_b in monolayer graphene acquired at 2.2 K (**a**), 100 K (**b**) and 200 K (**c**). The fast-scan axis is V_t , and the slow-scan axis is V_b . **d**, Arrhenius plot of saturation itinerant density n_H^s as a function of temperature. **e**, Replotting $P_{2D} - E$ hysteresis loops measured at 100 K, 200 K and 300 K for $|E|_{\max} = 36$ mV nm^{-1} . The slopes of each individual parallelogram loop are labeled as color dashed lines, corresponding to GSAS regions.

Here's a brief text followed by a question based on your provided content:

4. Your findings of electronic ferroelectricity in monolayer graphene not only enhance its remarkable properties but also open new avenues for exploring novel physics. The interaction of graphene's ferroelectricity with other characteristics, such as ferromagnetism, topology, and superconductivity, may lead to the emergence of unconventional multiferroics and topological ferroelectrics. From an application standpoint, monolayer graphene offers significant advantages for non-volatile memory and synaptic devices, including high mobility, stability, and multifunctionality. Notably, the ferroelectric features in graphene moiré superlattices can persist up to room temperature, making them suitable for advanced applications. Furthermore, the established methods for growing scalable monolayer graphene superlattices simplify the design of ferroelectric devices, as the required h-BN serves as both a substrate and dielectric material.

In your opinion, what advantages does monolayer graphene offer for non-volatile memory and synaptic devices compared to traditional materials?

Our response: We appreciate your excellent perspective. Monolayer graphene possesses various properties, including air stability, mechanic stiffness and high electron mobility. In the following, we list several advantages of graphene-based non-volatile memory, synaptic device and other

novel nanoelectronics.

1. Compared to traditional insulating ferroelectrics and semiconducting 2D ferroelectrics (e.g., 3R-MoS₂), graphene has ultrahigh electron mobility, which allows us to build ferroelectric transistors with faster switching and low-power consumption.
2. The coexistence of electronic ferroelectricity and other quantum states, such as correlated and topological phases, in graphene moiré materials endow graphene tremendous advantages for exploring new physics and designing novel quantum devices. For example, the interplay of electronic ferroelectricity with superconductivity and correlated Chern insulators can lead to the observations of ferroelectric superconductivity (Ref: *Nat. Nano.* 18, 331 (2023)) and ferroelectric Chern insulator (Ref: *Nat. Nano.* 19, 962 (2024)), respectively.
3. Intriguingly, we have observed multiple spontaneous polarization states, which can be continuously switched by electric fields. This multi-state switching is significantly different from traditional ferroelectrics, where only two states can be switchable. This unique switching capability enables graphene to show promising applications in multi-state memory and neuromorphic computing.
4. In traditional lattice-driven ferroelectric devices, the performance will degrade as the number of polarization-reversal cycles increasing. This phenomenon is also known as ferroelectric fatigue due to the migration of ionic defects. In contrast, the polarization switching in graphene ferroelectrics is purely electronic dynamic. Therefore, we can realize fatigue-free ferroelectric switching in graphene devices with much higher endurance.

In summary, graphene ferroelectrics possess unique advantages in terms of high mobility, high endurance, and multifunctionality compared to traditional ferroelectrics.

5. In your study, we investigated gate hysteresis in graphene devices, particularly its dependence on extrinsic effects such as charge traps from defects, impurities, or adsorbates. Unlike typical behavior, we observed a robust gate hysteresis that remained consistent across different scan rates and cycles. When fixing V_t at 0 V and varying V_b at scan rates from 150 mV s⁻¹ to 400 mV s⁻¹, the forward and backward scan curves displayed nearly identical resistance peak positions and magnitudes. Additionally, tests for endurance showed that repeated forward and backward scans of V_b or V_t for 20 cycles resulted in overlapping curves, indicating excellent cycle endurance. This robust hysteresis was also observed in monolayer, bilayer, and trilayer graphene moiré superlattices, suggesting a common underlying mechanism.

What factors contribute to the observed robustness of gate hysteresis in graphene devices, despite the influence of extrinsic effects like charge traps? and how does the endurance of gate hysteresis across multiple scan cycles reflect on the potential applications of monolayer, bilayer, and trilayer graphene moiré superlattices?

Our response: Thank you for raising this comment. We apologize for the confusion. In our devices, we didn't observe extrinsic effects contributing to gate hysteresis. This is reasonable because our samples were all h-BN encapsulated. Instead, we have several lines of evidence indicating the hysteresis is intrinsic, originating from electronic ferroelectricity.

First, the temperature dependences of GSAS and P_r are quite different from those caused by extrinsic effects, as we just discussed in Fig. R6.

Second, we have performed the scan-rate dependence of gate hysteresis. In the mechanisms dominated by extrinsic effect (such as charge traps), as illustrated in Fig. R7a, the hysteresis is highly dependent on the scanning rate, due to the existence of mobile ions or dipole moment in the system. In contrast, in our monolayer graphene superlattice devices, the measurements of longitudinal R_{xx} by forward and backward sweeps of V_b at fixed $V_t = 0$ with different scanning rates demonstrate that the hysteresis is independent of the scanning rate in terms of the resistance peak position.

Third, in the revised manuscript, we further performed the endurance measurements by switching the polarization states more than 10^4 times (limited by the instrument) as shown in Fig. R8. It's noted that these data were acquired six months after the initial measurements. Typically, charge traps will lead to performance degradation over several cycles. In contrast, as shown in Fig R8, our device does not show a substantial fatigue effect after 3×10^4 switching cycles. Our new results not only provide additional evidence for excluding the extrinsic effects, but also highlight the potential memory applications of graphene moiré superlattices with high endurance.

In the revised manuscript, we have added the new data and discussion in the Supplementary Note 3.

Fig. R7 | **a**, Schematic of charge-trapping hysteresis in a typical graphene device. From left to right, hysteresis becomes more pronounced with increasing scanning rate (Ref: *ACS Nano* 4, 7221-7228 (2010)). **b**, Hysteretic R_{xx} as a function of V_b at fixed $V_t = 0$ with scanning rate of 150, 300 and 400 mV/s in our monolayer graphene superlattice devices.

Fig. R8 | Endurance of graphene electronic ferroelectricity against switching cycles. Channel current I_{ds} of graphene moiré device measured with constant source-drain voltage V_{ds} after a series of positive and negative pulse voltages applied to the top gate. For each cycle, we applied ± 9 V pulse with a width of 0.6 s. After 10^4 cycles (the largest allowed cycles in our equipment), we repeated them for another 2 rounds. Within the whole test process, the polarization states were highly stable.

In general, the manuscript has a correct methodological structure. But for my opinion the authors should making a major revision on this manuscript. I hope that authors reconsider these all points, and they clarify some issues.

Our response: We thank you for your positive assessment of our work and raising constructive suggestions. We hope our point-to-point responses have addressed your concerns.

Reviewer #3 (Remarks to the Author):

In the manuscript "Electric ferroelectricity in monolayer graphene moire superlattices" the authors investigate the possibility to introduce ferroelectricity in graphene, both single-layer and few layer, by encapsulating it with h-BN. Their results are confirmed for 5 devices, whose geometry is given in Table S1 of Supplementary Information. The results sound interesting, although they lack an explanation of the microscopic mechanism leading to the observed phenomena.

Our response: We thank you for appreciating the merits of our work and for your insightful comments.

From the point of view of symmetry, it is known that the out-of-plane polarisation in a material or a heterostructure can be induced only in the case of a structural asymmetry in the z-direction. However, in one the devices analysed (D2-1), it is clear that the system possesses the horizontal mirror symmetry, since Theta_BN (the relative angle between the top and the bottom h-BN layer) is equal to zero. Thus, it is for me unclear how in such a configuration the polarisation can arise. It would be beneficial if the authors explain whether the top and bottom h-BN are equal and if yes, give an appropriate explanation of the phenomenon measured.

Our response: While the microscopic picture in the main text is described based on Device D1-1, the behavior in Device D2-1 can also be understood within the "asymmetric potential" framework.

1. Although the monolayer graphene is roughly aligned with both h-BN, the moiré potentials at

top and bottom interfaces are different in terms of the moiré wavelength. In other words, the twist angle between top h-BN and graphene is slightly different from that between bottom h-BN and graphene. As shown in Fig. R9c, there are two resistance peaks located at $V_{b1} = -15$ and $V_{b2} = -47$ V in the hole side, which correspond to two moiré wavelengths (*Sci Adv* 5, eaay8897 (2019)) arising from top and bottom interfaces, respectively. The different moiré sizes in top and bottom interfaces can result in their different moiré trapping ability. Thus, the horizontal mirror symmetry is absent in Device D2-1.

2. The gate response between V_t and V_b are dramatically different in detail. In the dual-gate resistance mappings, we scan V_t forward and backward between -20 V to 20 V at each fixed V_b . Intriguingly, V_t is completely screened within the scanning range, resembling the behavior of the main device. We then measured R_{xx} by scanning V_b at a fixed $V_t = 0$ V. Hysteresis behavior emerged when $|V_b|_{\max} > 30$ V, identified by the non-overlapped resistance peaks of SDP on the hole side. Our data indicate that the top gate has stronger moiré trapping ability than bottom gate, exhibiting top gate anomalous screening effects. Similar behavior has been observed in bilayer graphene aligned with both top and bottom h-BN (*Nature* 588, 71-76, 2020).

In conclusion, the interfaces at the top and bottom h-BN are not equivalent in Device D2-1. The asymmetric moiré potential at the two interfaces of graphene plays an important role in its ferroelectric behaviors.

In the revised manuscript, we have added above discussion in Supplementary Note 6.

Fig. R9 | **a,b**, Dual-gate maps of R_{xx} by scanning V_t forward (**a**) and backward (**b**) at each fixed V_b . **c**, Hysteresis dependence on scan range of V_b at fixed $V_t = 0$.

Point-by-point response to the Reviewers

We sincerely thank the reviewers for taking the time to assess our manuscript and raising constructive suggestions to improve it. We believe that this letter and the revised manuscript fully addressed your questions and comments. In the following, we respond to the comments in detail. Changes made in the updated manuscript are highlighted in yellow.

Reviewer #1 (Remarks to the Author):

Dear Authors,

After evaluating your revised manuscript, I am content with the provided responses to my previous concerns.

No more comments from my side.

Our response: We thank you for your constructive comments and recommending the publication of our work.

Reviewer #2 (Remarks to the Author):

The authors are answer to my questions and are improved their manuscript in my opinion this work merits the publications in Nature Communications.

Our response: We thank you for your constructive comments and recommending the publication of our work.

Reviewer #3 (Remarks to the Author):

I thank the reader for the answers provided, especially for pointing to the importance of the asymmetric potential.

I would have to disagree with the authors that the device D2-1 does not have the horizontal mirror symmetry. To prove this, we can set the $z=0$ to discuss the position of the graphene monolayer. In this case, the presence of horizontal mirror symmetry implies $f(z)=f(-z)$, meaning that the twist angle between the top hBN and the bottom hBN (θ_{BN}) has to be equal. This is written in Table S1., where Θ_{BN} is defined as the relative angle between the top h-BN and bottom h-BN determined by optical images and SHG measurements.

Our response: We thank you for giving us this opportunity to clarify the sample structure in our Device D2-1. We apologize for the confusion. The optical images and SHG measurements only provide a rough angle configuration, while the transport data can determine the precise angles between the top and bottom h-BN. In our SHG measurements, the angle resolution (limited by step size of the rotator) is only 5 degrees. With the best fitting result, we can determine the relative angle Θ_{BN} is roughly around 0 degree with an uncertainty of 5 degrees. From the optical images, we used straight edges of graphene and h-BN to estimate their angle configuration. This method is even more imprecise for determining the angle, as it's completely measured by a ruler. In the initial version, we present SHG and optical images in order to distinguish Device D2 from Device D1.

To precisely determine the angles between graphene and the two h-BN, we directly resort to the transport data. The small twist angle between graphene and h-BN can result in the formation of moiré superlattices.

The moiré unit cell area is given by $A = (\frac{\sqrt{3}}{2})\lambda^2$, where $\lambda = \frac{(1+\delta)a_G}{\sqrt{2(1+\delta)(1-\cos\theta)+\delta^2}}$ is the wavelength of the moiré pattern, $a_G = 0.246$ nm is the in-plane lattice constant of graphene, $\delta \approx 1.6\%$ is the lattice mismatch between graphene and h-BN, and θ is the twist angle between graphene and h-

BN. Since electrons in graphene has four-fold degeneracy (2 spins +2 valleys), to fully fill one unit cell, we need four electrons. Therefore, we have the relation of $A = 4/n_s$, where n_s corresponds to the carrier density at full fillings of a band. Typically, in the transport data, at full fillings of a moiré band, there will be a resistance peak because of minimum DOS. In Fig. R1a, we plot the longitudinal resistance R_{xx} as a function of carrier density n in Device D2-1, from which we can observe two different resistance peaks at hole sides. The resistance peak located at $n_{s1} = -2.75 \times 10^{12} \text{ cm}^{-2}$ is associated with the moiré superlattice formed between graphene and top h-BN. Apart from that, the resistance peak located at $n_{s2} = -6.34 \times 10^{12} \text{ cm}^{-2}$ corresponds to the moiré superlattice formed between graphene and bottom h-BN. Therefore, we can quantitatively determine the moiré wavelengths $\lambda_t = 13 \text{ nm}$, $\lambda_b = 8.6 \text{ nm}$ and the corresponding two twist angles $\theta_t = 0.58^\circ$, $\theta_b = 1.37^\circ$. Since the moiré potential is sensitive to the moiré wavelength, even slight difference in twist angles between graphene/top h-BN and graphene/bottom h-BN can significantly affect the moiré trapping ability, resulting in the asymmetric potential. With these results, we can further determine the relative angle between the top h-BN and bottom h-BN θ_{BN} is either 1.95° or 0.79° as illustrated in Fig. R1c. From this analysis, we think the structure symmetry is absent in Device D2-1.

In the revised manuscript, we have added the discussion in Supplementary Note 6 and the precise twist angle θ_b to the Supplementary Table 1.

Fig. R1 | **a**, Longitudinal resistance R_{xx} as a function of carrier density n . Top axis is the gate voltage of back gate. Both V_b and n are offset to zero. **b**, Moiré wavelength (black) and carrier density of fully filling moiré band (red) as a function of the angle between the graphene and h-BN. **c**, Two possible the lattice configuration illustrated with twist angles θ_t and θ_b between graphene and top/bottom h-BN. Both of which exhibit a nonzero twist angle between top h-BN and bottom h-BN.

Additionally, in the suggested reference Nature 588, 71–76 (2020) it is stated that "in the presence of ferroelectric ordering, the interlayer displacement field has two contributions, the field induced by gating and the internal electric field arising from the built-in ferroelectric polarization."

Have the authors distinguished the contribution of the interlayer displacement field due to gating and the built-in ferroelectric polarization? For me it is clear that the device can show polarization due to different top and bottom gating potentials but this does not imply that the sample has built-in ferroelectric

polarization, since the asymmetric potential can come solely due to gating.

Our response: We thank you for this fruitful comment. We indeed observed the built-in ferroelectric polarization due to the formation of electron-hole dipole moments.

In conventional ferroelectric materials, the built-in ferroelectric polarization arises from the spatial separation of the cation and anion, which can be unveiled by the $P - E$ hysteresis loop measurements. Similarly, in our system, we can reveal the built-in ferroelectric polarization from the measurements of $P - E$ loops. The difference is that the ferroelectricity in our system is purely driven by electron dynamics. Therefore, we can directly measure the polarization $P = en_{\text{H}}d_{\text{dipole}}$ from Hall density measurements. Furthermore, by using the dual-gate structure, we can independently control the n_{total} and E through the application of V_{t} and V_{b} with the relations given by $n_{\text{total}} = (C_{\text{b}}V_{\text{b}} + C_{\text{t}}V_{\text{t}})/e$ and $E = D/\epsilon_{\text{BN}} = (C_{\text{b}}V_{\text{b}} - C_{\text{t}}V_{\text{t}})/2\epsilon_0\epsilon_{\text{BN}}$. The full maps of $\Delta P = e\Delta n_{\text{H}}d_{\text{dipole}}$ as the function of n_{total} and E by measuring the hysteretic loops is shown in Fig. R2a. Specifically, when fixing $n_{\text{total}} = 0$, scanning E forward and backward can form a standard $P_{2\text{D}} - E$ hysteresis loop, as shown in Fig. R2b. Notably, at $n_{\text{total}} = 0$ and $E = 0$, which exactly corresponds to that both the top and bottom gates are zero ($V_{\text{b}} = 0$ and $V_{\text{t}} = 0$), we observed two opposite ferroelectric polarizations with $P_{2\text{D}} \neq 0$ as marked by black and red dots in Fig. R2b. The corresponding schematics are shown in Fig. R2c. Here, the electron-hole dipole moments are formed by the localized holes at top interface and itinerant electrons at bottom interface after a history of applying E . The bound of localized hole and itinerant electron pairs is dynamic, resulting in a built-in electric field.

Fig. R2 | **a**, The difference of Hall carrier density n_{H} between the forward and backward sweeps of external electric field E at each fixed carrier density n_{total} . **b**, Two-dimensional polarization $P_{2\text{D}}$ as a function of E measured by sweeping E sequentially in the direction denoted by the arrows at fixed $n_{\text{total}} = 0$. **c**, Schematic of built-in electric field and external electric field by gating.

I cannot recommend the manuscript for publication without adequately addressing these questions.

Our response: We hope our point-to-point responses have addressed your concerns.